# Silver and Copper Nanoparticles as the New Biocidal Agents Used in Pre- and Post-Milking Disinfectants with the Addition of Cosmetic Substrates in Dairy Cows

**DOI:** 10.3390/ijms24021658

**Published:** 2023-01-14

**Authors:** Aleksandra Kalińska, Sławomir Jaworski, Mateusz Wierzbicki, Magdalena Kot, Daniel Radzikowski, Sebastian Smulski, Marcin Gołębiewski

**Affiliations:** 1Animal Breeding Department, Warsaw University of Life Sciences, 02-786 Warszawa, Poland; 2Department of Nanobiotechnology, Warsaw University of Life Sciences, 02-786 Warszawa, Poland; 3Department of Internal Diseases and Diagnostics, Poznań University of Life Sciences, 60-637 Poznań, Poland

**Keywords:** mastitis, teat dipping, silver, copper, nanoparticles, dairy cows

## Abstract

*Mastitis* is one of the most common issues for milk producers around the world. Antibiotic therapy is often ineffective, and therefore, scientists must find a new solution. The aim of this paper is to estimate the influence of common and well-known cosmetic substrates and mixtures of nanoparticles (NPs) and cosmetic substrates on the viability of frequently occurring mastitis pathogens, *Escherichia coli* (*E. coli*) and *Staphylococcus aureus* (*S. aureus*). The obtained results suggest that only collagen + elastin and glycerine influenced and increased bacteria viability. In case of the rest of the cosmetic substrates, the viability of *E. coli* and *S. aureus* was decreased, and the results were statistically significant (*p* ≤ 0.01). Prepared pre-dipping and dipping mixtures decrease (*p* ≤ 0.01) the viability of the mentioned pathogens. The obtained results of the in vitro analysis are very promising. In the next step, prepared mixtures should be tested in different herd conditions if they can be used in mastitis prevention or decrease the number of subclinical mastitis cases. Furthermore, these mixtures could become an interesting alternative for organic milk production where conventional preparations and antibiotics are forbidden. However, further analysis, especially on the influence of prepared mixtures on other bacteria species and, algae, fungi, are necessary.

## 1. Introduction

Nowadays, nanotechnology is one of the most rapidly developing fields of technology and science. The possibility of producing materials, objects, and substances on a nanoscale level gives scientists new skills and tools [1] to fight, e.g., diseases. However, the use of NPs in udder disinfectant preparations is still limited, but some authors indicate their high potential in mastitis prevention [2]. Our previous studies revealed that AgNPs, CuNPs, and AgCuNPs are non-toxic for human and bovine mammary gland cells and decrease the viability *E. coli* and *S. aureus* [3]. Similar data were also presented in previous papers [4,5,6,7,8,9,10]. Moreover, the combination of AgNPs and CuNPs can decrease pathogens viability and has high biocidal properties that can also reduce bacterial biofilm by nearly 100% [11]. Therefore, many papers revealed that NPs can be used as new agents in alternative therapies to treat bovine mastitis [3,11].

One of the biggest concerns in case of nanoparticles is their persistence in the environment. The toxic influence of AgNPs and CuNPs is mainly observed in water environments [12]; however, some researchers point out that CuNPs are more toxic than AgNPs [13]. The exposure to NPs can cause several negative changes in the aquatic organisms, e.g., increased oxidative stress, reproductive disorders or impairing activity of the immune system [12]. On the other hand, there are several experiments proving that the effect of NPs on living organisms can be beneficial. For example, the in ovo inoculation of calcium carbonate nanoparticles on chicken embryo influence bone quality by improving their mineralization if the process is conducted during embryogenesis [14]. This phenomena are not completely understood; therefore, further studies are necessary to document the long-term influence of NPs on living organisms.

In general, disinfectant preparations used as pre-dipping or dipping solutions have an antibacterial effect, but frequently, they do not positively influence udder skin. Moreover, they can dry udder skin and as a consequence cause secondary skin problems. On the other hand, effective mastitis prevention should include disinfectant preparation because of their antibacterial activity but also to protect the teat canal from closing for at least 30 min after milking. This period of time is longer in older cows.

Nowadays, many moisturizing and nutritional skincare components are available in the marketplace. Although the list of components may be similar, differences in the emulsion properties of the final products provide different therapeutic efficacy [15]. Modern products contain different moisturizing agents such as humectants or occlusives. Humectants, e.g., glycerin, propylene glycol, sodium lactate, urea, vitamins, proteins, etc., can help the skin retain moisture and promote flexibility of the skin. Occlusives, e.g., petrolatum, lanolin alcohols, jojoba oil, cocoa butter, paraffin, cholesterol, heavy lipid mixtures, olive oil, and heavy mineral oil create a barrier on the skin that allows the skin to maintain its own and natural moisture [15].

However, the number of agents used in skincare products is much larger. They can include products such as vitamins A, E, D, and B, retinoids, linoleic acid, biotin, or flower and herbal extracts. On the other hand, these substances must be present in the product at the sufficient concentration to positively affect the skin physiology [15].

Therefore, the authors decided to select common, inexpensive, and widely available cosmetic substrates and evaluate whether they can be used in mastitis disinfectant preparations. The chosen cosmetic substrates were collagen + elastin (1%), glycerine (5%), sorbitol (5%), propylene glycol (8%), d-panthenol (1.3%), vitamin C (5%), sodium lactate (5%), urea (5%), and marigold flower extract (5%).

The first aim of the study was the in vitro analysis of the influence of cosmetic substrates and the impact of both, NPs and cosmetic substrates, on the viability of *mastitis* pathogens: *E. coli* and *S. aureus*. In the second part of the study, the viability of bacteria was calculated after 24 h incubation in two pre-dipping (Pre1 and Pre2) and dipping (Dipp1 and Dipp2) mixtures containing AgCuNPs and cosmetic substrates.

## 2. Results

### 2.1. The In Vitro Viability of Mastitis Bacteria Species after Incubation with Cosmetic Substrates

The viability of *E. coli* and *S. aureus* cells after incubation with commercially available cosmetic substrates and an addition of 1 ppm NPs is presented in Table 1 and Table 2. The number of bacteria cells in the control group was estimated as 100%. The obtained results from each group were presented as a percentage of the control group. The viability of selected bacteria species decreased (*p* ≤ 0.01) by 20–30% after incubation with only AgNPs, CuNPs, or AgCuNPs.

In general, the viability of bacteria species decreased if they were incubated with NPs, cosmetic substrates, or both. The exceptions for *E. coli* were media with the addition of collagen + elastin (122.74%), AgNPs + ce (104.75%), glycerine (146.69%), AgNPs + g (101.89%), CuNPs + g (122.97%), AgCuNPs + g (107.33%), and AgNPs + mfe (134.66%). For *S. aureus*, increased viability was observed in similar groups: glycerine (130.68%), AgNPs + g (111.18%), CuNPs + g (124.88%), AgCuNPs + g (110.36%), and in the group containing an addition of collagen + elastin: CuNPs + ce (116.26%), AgCuNPs + ce (105.72%).

The in vitro analysis revealed that for collagen + elastin, propylene glycol, urea, and d-panthenol in the *E. coli* group, the viability of bacteria cells decreased (*p* ≤ 0.01) after incubation with all NPs compared to the control group. Statistical differences within groups (*p* ≤ 0.01) occurred in media with the addition of collagen + elastin, d-pantenol, urea, and marigold flower extract. The addition of sorbitol, glycol propylene and urea generally decreased the viability of bacteria by 50–60%. In case of collagen + elastin, lower viability than in the control group was observed in groups CuNPs + ce (*p* ≤ 0.05) and AgCuNPs + ce (*p* ≤ 0.01). The usage of d-panthenol decreased bacteria viability (*p* ≤ 0.01). However, the lower viability (31.19%) was observed in group containing only this substrate without NPs. In all groups with vitamin C, the viability of *E. coli* was decreased (*p* ≤ 0.01) compared to the control group. In the experimental group with the addition of glycerine, statistical differences occurred only in comparison with CuNPs (*p* ≤ 0.01). However, the viability of bacteria in all flasks with medium containing glycerine addition was higher than in control group (by 10–45%). Only in the group with the addition of sodium lactate were no statistical differences observed.

The results of in vitro analysis of the influence of glycerine, propylene glycol, vitamin C, sodium lactate, urea, and marigold flower extract on the viability of *S. aureus* were similar to *E. coli* groups. However, in case of *S. aureus*, statistical differences occurred in all groups (*p* ≤ 0.01). The addition of glycerine influenced an increase in bacteria viability compared to the control group (*p* ≤ 0.01). The usage of sorbitol, propylene glycol, sodium lactate and urea decreased the viability of *S. aureus* by 20% (*p* ≤ 0.05) to even 50% (*p* ≤ 0.01). The strongest effect was observed in the vitamin C group, and the decrease (*p* ≤ 0.01) was similar within the group (around 65%).

### 2.2. The In Vitro Viability of Mastitis Bacteria Species after Incubation with Evaluated Pre-Dipping and Dipping Mixtures

The in vitro viability of *E. coli* and *S. aureus* cells after incubation in prepared pre-dipping and dipping mixtures was presented as a percentage of the control group, which was estimated as 100% (Table 3). 

The general phenomenon in the *E. coli* group was decreased viability in Pre1, Pre2, Dipp1, and Dipp2 compared to the control group and mixtures with the addition of NPs. This decrease was statistically significant (*p* ≤ 0.01) between the control group and the mixtures containing only NPs as well as all pre-dipping and dipping mixtures. For Pre1 and Pre2, a decrease in viability compared to the mixture with AgNPs was also observed.

For *S. aureus*, a statistically significant decrease in viability was observed between the control group and the mixtures with the additions of NPs and Dipp2. However, a decrease in in vitro viability was observed after incubation with all prepared mixtures, and this difference in viability was significant (*p* ≤ 0.05) between the mixtures with NPs and Pre1 and Pre2.

## 3. Discussion

Disinfectant pre-dipping and dipping preparations should mix antibacterial and udder skincare properties. The authors decided to estimate the incubation time for 24 h even though the pre-dipping and dipping products have contact with teat skin for 30–40 s or 8–10 h, respectively. However, the important aspect of the conducted experiments was the fact that selected cosmetic substrates could not increase the bacteria multiplication. Therefore, the authors decided to extend incubation time. 

Cosmetic substrates were chosen to complement each other and affect udder skin in different ways: moisturize, nourish, create biofilm and present preservative effect. The substrates included collagen + elastin (1%), glycerine (5%), sorbitol (5%), propylene glycol (8%), d-panthenol (1.3%), vitamin C (5%), sodium lactate (5%), urea (5%), and marigold flower extract (5%). Glycerine, sodium lactate, and urea have a strong moisturizing effect and are basic components of numerous skin-care cosmetics in different brands [15]. Collagen and elastin are used together because their properties complement each other and have regenerating and rejuvenating effects, restore freshness, smoothness, and elasticity of the skin [16]. Another important feature of soluble collagen is the fact that it creates a thin layer on the skin—a biofilm that protects tissue against environmental pollutions and excessive water loss. Propylene glycol has a positive impact on other cosmetic substrates and is a promotor for them to penetrate skin tissue, but it also can be treated as a preservative [17]. D-panthenol is a provitamin B5 and can easily penetrate deep skin layers, hair, and nails. It penetrates the epidermis, accumulates in the dermis, and changes into vitamin B5 or pantothenic acid. D-panthenol applied externally acts on the skin—it accelerates the division of skin cells and soothes irritated and red skin [18,19]. The basic and biologically active form of vitamin C is ascorbic acid. Vitamin C stimulates better blood supply to the skin by supporting the collagen synthesis process, but it also improves the firmness and elasticity of the skin and strengthens the walls of blood vessels [20]. Marigold flower extract is rich in flavonoids and carotenes. It accelerates wound healing, soothes irritations, prevents flaking, and has anti-inflammatory, antibacterial, and antifungal properties [21].

Good hygiene of milking routine is a key factor that can reduce the number of *mastitis* pathogens in a herd and decrease the rate of udder inflammations [22]. Unfortunately, using pre-dipping and dipping solutions is still not as popular as it should be, especially in small farms. One of the main factors affecting milk producers’ decisions regarding udder disinfectant preparations is the price. However, farmers should mainly focus on the composition of the solution. It should be balanced between antibacterial and antifungal properties and its influence on udder skin [22]. 

Previous papers have revealed that *mastitis* is a multifactorial disease. Several factors can contribute to maintaining a low infection rate in non-disinfected teats, e.g., low herd somatic cell count (SCC), poor herd parity, a high standard of the parlor and a clean environment, systematic liner changes, and regular cow tail clipping. Furthermore, the application of disinfectant preparations to uncleaned teats may affect the effectiveness of the products [23].

The fact that udder cleaning before milking and manual drying with a clean towel decreases the number of bacteria was described in the 1980s [24,25]. In herds with high infection rates and the risk of infection spread, the pre-milking teat disinfection of clean teats may bring benefits if followed by teat drying. However, some authors suggest that routine pre-milking teat disinfection in pasture-grazed herds will probably not be beneficial in herds with an SCC below 200 × 10^3^ cells/mL [23]. It should also be pointed out that pre-dipping can be the most effective way of reducing the amount of environmental bacteria, e.g., *E. coli* or *Streptococcus uberis* [26]. However, the same author indicated that post-milking teat dipping is crucial in *mastitis* management and prevention. Proper dipping coats each teat thoroughly in dipping solution and should be routine during lactation [23]. Good pre-dipping disinfectants also have another valuable trait—they help with the formation of the keratin plug in the teat canal after milking. The proper formation of the keratin plug is necessary to close the teat canal between each milking and reduce the probability of bacterial penetration of the teat canal [22]. The comparison of different forms of dipping application suggests that coating is better than spraying. It requires less solution and is more accurate. However, dipping preparations should always be used according to producers’ recommendations [26].

Improvement in *mastitis* prevention is necessary. New substances, methods, and alternative and natural solutions should be the main interest of scientists as well as their major challenge, as the inappropriate and excessive use of antibiotics in *mastitis* treatment for dairy cows has contributed to the increased antibiotic resistance of several mastitis pathogens [27].

Metal NPs are a potential alternative in mastitis prevention and the treatment of subclinical cases [3,11]. An interesting idea could also be the usage of propolis and honey. The authors of the study revealed that propolis presented higher antibacterial activity against *S. aureus* in comparison with honey. In the same research, *E. coli* was less susceptible to both honey and propolis [28].

Previous papers have presented the influence of AgNPs on the antimicrobial activities of common antibiotics against *S. aureus* and *E. coli*. The authors suggested that the combination of penicillin G, amoxicillin, erythromycin, clindamycin, and vancomycin together with AgNPs caused an increased activity of the mentioned drugs against both strains. The strongest enhancing effects of AgNPs were observed if vancomycin, amoxicillin, and penicillin G were used against *S. aureus* [29].

Results obtained in this study revealed that the viability of *E. coli* and *S. aureus* decreased if NPs, or most of the cosmetic substrates, or both were present in the mixtures. Previous papers have also reported that AgNPs and CuNPs have antibacterial effects on the pathogens studied in this article [3,30,31,32]. Furthermore, some authors also suggest that the combination of AgNPs and CuNPs and their synergistic effect should have a more complete bactericidal effect against mixed bacteria species [3,33].

In some groups, the selected cosmetic substrates could influence the increase in the number of bacteria. The increased viability of *E. coli* and *S. aureus* in mixtures with an addition of glycerine may occur because glycerine is used by bacteria as an effective nutrition component that results in a higher number of bacteria. The higher number of *E. coli* and *S. aureus* bacteria observed in collagen + elastin and marigold flower extract could be connected with different and random mistakes during analysis or just errors in pipetting. However, collagen and elastin are valuable proteins and excellent sources of amino acids that could be used by bacteria in the replication process. It is well known that these substrates are a valuable component of antiaging human cosmetic and are often used in aesthetic medicine treatments in order to improve skin condition [18,19]. The antimicrobial effect of propylene glycol has been presented in previous studies [34,35]. This substrate is also an effective preservative. In this study, sodium lactate also showed effective antimicrobial activity and decreased bacterial viability by around 30%. These results confirm previous findings about its antibacterial effect [36,37]. Vitamin C also seems to be an effective substrate that can have toxic impact on some bacteria. It is well known that vitamin C is a strong antioxidant. However, the in vitro research evaluating the influence of vitamin C on *E. coli* and *S. aureus* has already been examined [37]. These results suggest that vitamin C especially affects the metabolism of *S. aureus* and is likely to result in growth inhibition. The aerobic metabolism of vitamin C increases oxidative stress on bacterial cells, and therefore, vitamin C may be a safe and natural alternative if *S. aureus* growth must be inhibited in non-toxic conditions [38]. The antimicrobial influence of vitamin C on *E. coli* was stronger than its influence on *S. aureus* in this paper. The viability of the bacteria was 22.02% and 35.64%, respectively. The combination of glycol propylene and vitamin C should be an effective and natural preservative for the designed mixtures if they will be commercialized.

Prepared mixtures of NPs with cosmetic substrates were prepared on the basis of the viability results explained above. The combination of cosmetic substrates and NPs decreased the viability of *E. coli* and *S. aureus* by around 30–50% in each preparation (Pre1, Pre2, Dipp1, Dipp2). Presented data are also a promising perspective for further analysis, because in general, the number of bacteria in herd conditions should be lower and NPs should have more a favorable environment for their way of action. These results suggest that the formulated mixtures negatively influence bacteria viability and therefore could be used in mastitis prevention and maybe in the treatment of subclinical cases. The mixtures contain antibacterial substances, but also should have a positive impact on udder skin because they contain moisturizing components and create a protective biofilm on the skin. The experiment described in this paper included data only from the in vitro analysis. Therefore, another step should be the experiment in different herd conditions. Moreover, one of the major advantages of the prepared mixtures is fact that they do not contain parabens, silicones, or preservatives that are generally considered as harmful, e.g., by consumers buying human skin care products. These mixtures could possibly become an interesting solution especially in ecological farms where conventional treatments are forbidden or require a long waiting period after the antibiotic therapy. Pre-dipping and dipping mixtures containing substrates from natural sources should be a very promising innovation for ecological farms. However, the main key in mastitis prevention and management is always regularity and consistent compliance with hygiene rules. In addition, all mixtures have been reported to the Polish Patent Office.

## 4. Materials and Methods

### 4.1. Preparation of AgCuNPs through Self-Organization Phenomenon

The AgCuNP solution was prepared by mixing hydrocolloids of AgNPs at a concentration of 50 mg/L (Nano-Tech, Warsaw, Poland) and CuNPs at a concentration of 50 mg/L (Nano-Tech), using a 1:1 ratio. The prepared AgCuNP solution was incubated for 24 h at a temperature of 24 °C and then used in further analysis.

### 4.2. Bacterial Cultures

The bacteria of *E. coli* and *S. aureus* were obtained from LGC Standards (Łomianki, Poland) and kept in a 20% glycerol solution at −20 °C.

Bacteria cells were thawed and rinsed with sterile distilled water in order to remove glycerol. In the next step, bacterial cells were added to a nutrient broth medium (Bio-Rad, Warsaw, Poland) that was previously sterilized in glass flasks in an autoclave (Classic 2100, Prestige Medical, Chesterfield, UK). Flasks were placed in a rotating incubator at a temperature of 37 °C (SI500) for 24 h with a speed rotation of 70 rounds per hour.

### 4.3. Influence of Cosmetic Substrates on Bacterial Viability

Bacterial cells pipetted from the cell culture and incubated for one night at 37 °C were used. Experimental groups contained nutrient broth (Biomaxima, Lublin, Poland) and AgNPs, CuNPs, or AgCuNPs at concentrations of 1 ppm as well as one of the common cosmetic substrates. The cosmetic substrates were collagen + elastin (1%), glycerine (5%), sorbitol (5.4%), propylene glycol (8%), d-panthenol (1.3%), vitamin C (5%), sodium lactate (5%), urea (5%), and marigold flower extract (5%) (zrobsobiekrem.pl (accessed on 12 December 2022), Poland). The concentrations were chosen according to manufacturer’s suggestion (www.zrobsobiekrem.pl (accessed on 12 December 2022), Poland) and the author’s preliminary tests (not included in the paper) in order to achieve the most effective activity of each substrate. The control group was nutrient broth without the addition of NPs or any cosmetic substrate. Each group was prepared in triplicate. 

In the next step, 100 μL of the microorganism species mentioned in previous sub-section was added to prepared flasks. Samples were incubated for 24 h in a rotating incubator (SI500) at a temperature of 37 °C with a rotation speed of 70 rounds per minute. The viability of microorganisms was calculated using a PrestoBlue test. After incubation, 90 µL of medium was placed in 96-well plates, and 10 µL of PrestoBlue reagent (ThermoFisher Scientific, Warszawa, Poland) was added to each well. Each sample was placed in the plate in six repetitions. Plates were incubated for 20 min at 37 °C. Absorbance was measured using a wavelength of 570 nm in an immunoenzymatic Infinite M200 reader (Tecan, Durham, NC, USA). The only exception was vitamin C, because it immediately breaks down the PrestoBlue reagent. Results for this group are presented using an XTT kit and calculated according to the producer’s instruction. 

The viability of pathogens was presented as a percentage of the viability of the control group, according to the following equation:X = (optical sample density × 100%)/optical control group densityX = pathogen viability.(1)

### 4.4. Influence of Pre-Dipping and Dipping Mixtures on Bacterial Viability

Bacterial cells pipetted from the cell culture and incubated for one night at 37 °C were used. Experimental groups contained nutrient broth and AgNPs, CuNPs, or AgCuNPs at concentrations of 1 ppm as well as one of the preliminary mixtures of cosmetic substrates. Prepared mixtures contained: Pre1—vitamin C 2%, glycerine 2%, d-panthenol 1%, propylene glycol 8%, urea 5%, sodium lactate 5%; Pre2—vitamin C 2%, glycerine 1,5%, d-panthenol 1%, propylene glycol 8%, urea 5,5%, sodium lactate 5%; Dipp1—vitamin C 2%, glycerine 2%, collagen + elastin 1%, d-panthenol 1%, propylene glycol 8%, marigold flower extract 2%, sorbitol 5%, urea 5%; Dipp2—vitamin C 2%, glycerine 1%, collagen + elastin 1%, d-panthenol 1%, propylene glycol 8%, marigold flower extract 1%, sorbitol 4.2%, urea 5.4%. The control group was nutrient broth without the addition of NPs or any cosmetic substrate. Each group was prepared in triplicate. 

In the next step, 100 μL of each bacteria species was added to prepared flasks. Samples were incubated for 24 h in a rotating incubator (SI500) at a temperature of 37 °C with a rotation speed of 70 rounds per minute. The viability of microorganisms was calculated using an XTT kit. After incubation, 100 µL of mixture was placed in 96-well plates, and 50 µL of XTT reagents mix (ThermoFisher Scientific, Poland) was added to each well. Each sample was placed in the plate in six repetitions. Plates were incubated for 20 min at 37 °C. Absorbance was measured using a wavelength of 450 nm in an immunoenzymatic Infinite M200 reader (Tecan, Durham, NC, USA).

The viability of pathogens was presented as a percentage of the viability of the control group, according to the Equation (1) (Section 4.3).

### 4.5. Statistical Analysis

Statistical analysis was conducted in the IBM SPSS Statistics 24 program. One-way analysis of variance (ANOVA) was used to estimate if there were statistical differences between and within the groups.

## 5. Conclusions

In summary, the in vitro experiment revealed that the combination NPs and cosmetic substrates could be an effective combination against *S. aureus* and *E. coli* in mastitis prevention. The obtained results suggest that the combination of AgNPs, CuNPs, or AgCuNPs and common cosmetic substrates could be a new and innovative solution in mastitis prevention, because most of the commercially available preparations have mainly antibacterial function. Most of the selected substances do not positively influence bacteria growth during 24 h of incubation. Moreover, some of them, e.g., propylene, glycol, or vitamin C, can decrease the number of bacteria by 35–50%. The prepared pre-dipping and dipping mixtures of NPs and cosmetic substrates could also be an alternative in mastitis prevention in organic farms where antibiotics are strictly forbidden. However, further studies are necessary to evaluate the biocidal effect of designed mixtures against other mastitis pathogens and achieve higher biocidal properties of prepared mixtures.

## Figures and Tables

**Table 1 ijms-24-01658-t001:** The in vitro viability (%) of *Escherichia coli* cells after incubation with commercially available cosmetic substrates and NPs.

Control Group		Collagen + Elastin 1%		Glycerine 5%		Sorbitol 5.4%		Propylene Glycol 8%	
C	100.00 ^A^	collagen + elastin	122.74 ^ABCDEG^	glycerine	146.69 ^C^	sorbitol	59.95 ^AaD^	propylene glycol	47.66 ^ABCD^
AgNPs	79.20 ^AaB^	AgNPs + ce	104.75 ^aBCDEF^	AgNPs + g	101.89 ^C^	AgNPs + s	53.68 ^AB^	AgNPs + pg	50.05 ^ABCD^
CuNPs	82.66 ^AaC^	CuNPs + ce	98.64 ^acDG^	CuNPs + g	122.97 ^C^	CuNPs + s	53.76 ^AB^	CuNPs + pg	44.03 ^ABCD^
AgCuNPs	79.17 ^AaD^	AgCuNPs + ce	88.56 ^AcDF^	AgCuNPs + g	107.33 ^C^	AgCuNPs + s	55.72 ^AB^	AgCuNPs + pg	39.22 ^ABCD^
**D-panthenol 1.3%**		**Vitamin C 5%**		**Sodium Lactate 5%**		**Urea 5%**		**Marigold Flower Extract 5%**	
d-panthenol	31.19 ^ABCDJ^	vitamin C	22.02 ^A^	sodium lactate	74.62	urea	41.17 ^ABCD^	marigold flower extract	92.17 ^AHd^
AgNPs + dp	66.10 ^AJCK^	AgNPs + vit C	64.27 ^A^	AgNPs + sl	62.80	AgNPs + u	51.00 ^ABCDb^	AgNPs + mfe	134.66 ^ABCDHI^
CuNPs + dp	34.48 ^ABCDK^	CuNPs + vit C	82.38 ^A^	CuNPs + sl	75.06	CuNPs + u	43.01 ^ABCD^	CuNPs + mfe	69.12 ^AaDHI^
AgCuNPs + dp	47.85 ^AC^	AgCuNPs + vit C	74.95 ^A^	AgCuNPs + sl	48.80	AgCuNPs + u	39.82 ^ABCDb^	AgCuNPs + mfe	76.06 ^ADdI^

Differences significant at *p* ≤ 0.01 for A, B, C, D, E, F, G, H, I, J, K. Differences significant at *p* ≤ 0.05 for a, b, c, d.

**Table 2 ijms-24-01658-t002:** The in vitro viability (%) of *Staphylococcus aureus* cells after incubation with commercially available cosmetic substrates and NPs.

Control Group		Collagen + Elastin 1%		Glycerine 5%		Sorbitol 5.4%		Propylene Glycol 8%	
C	100.00 ^Ae^	collagen + elastin	87.71 ^i^	glycerine	130.68 ^ABCD^	sorbitol	87.57 ^Def^	propylene glycol	54.04 ^ABCD^
AgNPs	62.72 ^ABb^	AgNPs + ce	85.28 ^j^	AgNPs + g	111.18 ^AbD^	AgNPs + s	87.28 ^Deg^	AgNPs + pg	59.79 ^ABCD^
CuNPs	93.16 ^AaBCDg^	CuNPs + ce	116.26 ^ABgDij^	CuNPs + g	124.88 ^ABCD^	CuNPs + s	76.44 ^AbCD^	CuNPs + pg	55.61 ^ABCD^
AgCuNPs	85.67 ^ABCDh^	AgCuNPs + ce	105.72 ^Ah^	AgCuNPs + g	110.36 ^AbD^	AgCuNPs + s	75.12 ^AbCDfg^	AgCuNPs + pg	55.07 ^ABCD^
**D-panthenol 1.3%**		**Vitamin C 5%**		**Sodium Lactate 5%**		**Urea 5%**		**Marigold Flower** **Extract 5%**	
d-panthenol	88.93 ^ADd^	vitamin C	35.64 ^ABCD^	sodium lactate	66.11 ^ABCDG^	urea	52.07 ^ABCD^	marigold flower extract	90.13 ^ABCDI^
AgNPs + dp	97.88 ^bcdE^	AgNPs + vitC	35.10 ^ABCD^	AgNPs + ms	68.42 ^ABCDH^	AgNPs + u	56.40 ^ABCD^	AgNPs + mfe	65.23 ^ABCDIJ^
CuNPs + dp	98.40 ^bcdF^	CuNPs + vitC	35.40 ^ABCD^	CuNPs + ms	82.54 ^ABCDGH^	CuNPs + u	52.15 ^ABCD^	CuNPs + mfe	91.87 ^IJK^
AgCuNPs + dp	84.45 ^AaDEF^	AgCuNPs + vitC	35.81 ^ABCD^	AgCuNPs + ms	82.19 ^ABCDGH^	AgCuNPs + u	51.76 ^ABCD^	AgCuNPs + mfe	73.94 ^AghIK^

Differences significant at *p* ≤ 0.01 for A, B, C, D, E, F, G, H, I, J, K. Differences significant at *p* ≤ 0.05 for a, b, c, d, e, f, g, h, i, j

**Table 3 ijms-24-01658-t003:** The in vitro viability (%) of *E. coli* and *S. aureus* cells after incubation in prepared pre-dipping and dipping mixtures containing cosmetic substrates and NPs.

*E. coli*		*S. aureus*	
C	100 ^A^	C	100 ^Aa^
AgNPs	67.32 ^AaB^	AgNPs	30.51 ^Ab^
CuNPs	57.16 ^A^	CuNPs	28.14 ^Ac^
AgCuNPs	49.81 ^Aa^	AgCuNPs	26.75 ^Ad^
Pre1	48.58 ^AaB^	Pre1	73.95 ^bcd^
Pre2	42.47 ^ABc^	Pre2	74.11 ^bcd^
Dipp1	51.43 ^Ac^	Dipp1	47.23 ^a^
Dipp2	57.84 ^A^	Dipp2	44.32 ^A^

Differences significant at *p* ≤ 0.01 for A, B. Differences significant at *p* ≤ 0.05 for a, b, c, d.

## Data Availability

Data available on request.

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
