# Peer review of "Silver and Copper Nanoparticles as the New Biocidal Agents Used in Pre- and Post-Milking Disinfectants with the Addition of Cosmetic Substrates in Dairy Cows"

_ijms, 2023, doi:10.3390/ijms24021658_

Round 1

Reviewer 1 Report

1. The central idea of this article is the need to combine AgNPs, CuNPs, and AgCuNPs with cosmetic substrates of different ingredients to inhibit the common pathogens of dairy cow mastitis, Escherichia coli and Staphylococcus aureus, and the article deals with a large part of the discussion of cosmetic matrices, but the extremely important cosmetic matrix is not mentioned in the title of the article.

2. In vitro viability tests, certain cosmetic substrates exhibit much greater inhibitory effects on Escherichia coli and Staphylococcus aureus than AgNPs, CuNPs and AgCuNPs, such as propylene glycol, vitamin C and urea, these organic ingredients are more health-friendly to dairy cows than inorganic silver and copper, and the cost is lower and the source is more convenient, so additional AgNPs, CuNPs and AgCuNPs look a bit redundant.

3. In the in vitro viability test of E. coli, the control group showed that the mixture of AgNPs and CuNPs (AgCuNPs) had a synergistic strengthening effect on the inhibition of E. coli, but after combining with some cosmetic matrices with good bacteriostatic effect, this synergistic effect disappeared, such as sorbitol, vitamin C, which indicated that these cosmetic matrices had an inhibitory effect on the bactericidal properties of AgNPs and CuNPs. Adding AgNPs and CuNPs to cosmetic substrates that already have good bactericidal performance reduces the sterilization effect. In addition, there are also cases where the bactericidal effect of AgNPs and CuNPs is very different, such as d-panthenol, marigold flower extract, which indicates that these cosmetics have chemical reactions with AgNPs or CuNPs, and the impact of these chemical reactions on the quality of cows and milk is not discussed.

Author Response

Dear Reviewer,

Thank you for all valuable suggestions, all of them were included in the paper.

  1. The central idea of this article is the need to combine AgNPs, CuNPs, and AgCuNPs with cosmetic substrates of different ingredients to inhibit the common pathogens of dairy cow mastitis, Escherichia coli and Staphylococcus aureus, and the article deals with a large part of the discussion of cosmetic matrices, but the extremely important cosmetic matrix is not mentioned in the title of the article.

The addition of cosmetic substrates is important issue and should be included in the title. Therefore, title should be changed according to the received recommendations to fully reflect the case of the study and be more clear for future readers.

The selected bacteria species are well know and commonly used in modern studies. The idea of evaluation the effect of designed mixtures on different bacteria eg., streptococci, CNS and Corynebacterium sp and other mastitis pathogens eg., fungi, algae should be next step of in vitro analysis. Thank you for emphasizing this problem. The authors decided to choose E. coli and S. aureus also because these bacteria are commonly used in different fields of microbiological studies. Moreover, in Poland S.aureus is still common problem in many herds so the high effectiveness of disinfection products against these bacteria is crucial. Thank you for this advice, future research will include higher range of pathogen species. The conclusions were also improved, according to suggestion of the Reviewer.

  1. In vitro viability tests, certain cosmetic substrates exhibit much greater inhibitory effects on Escherichia coli and Staphylococcus aureus than AgNPs, CuNPs and AgCuNPs, such as propylene glycol, vitamin C and urea, these organic ingredients are more health-friendly to dairy cows than inorganic silver and copper, and the cost is lower and the source is more convenient, so additional AgNPs, CuNPs and AgCuNPs look a bit redundant.

The addition of NPs in case of organic farming could be redundant. It is very interesting idea to compare the biocidal effect of designed mixture without NPs. Moreover, in future studies authors will plan to used wider range of pathogen species to estimate the potential of proposed mixtures.

  1. In the in vitro viability test of E. coli, the control group showed that the mixture of AgNPs and CuNPs (AgCuNPs) had a synergistic strengthening effect on the inhibition of E. coli, but after combining with some cosmetic matrices with good bacteriostatic effect, this synergistic effect disappeared, such as sorbitol, vitamin C, which indicated that these cosmetic matrices had an inhibitory effect on the bactericidal properties of AgNPs and CuNPs. Adding AgNPs and CuNPs to cosmetic substrates that already have good bactericidal performance reduces the sterilization effect. In addition, there are also cases where the bactericidal effect of AgNPs and CuNPs is very different, such as d-panthenol, marigold flower extract, which indicates that these cosmetics have chemical reactions with AgNPs or CuNPs, and the impact of these chemical reactions on the quality of cows and milk is not discussed.

The authors did not decided to estimate the influence of designed mixtures because the experiment was conducted in vitro conditions. Further studies should include very careful evaluation of the potential effect on 1. milk quality, 2. udder health exposed to long term contact with NPs. However, previous studies carried out by authors suggest that NPs do not negatively influenced bovine mammary gland cells (https://doi.org/10.3390/ijms20071672). All these suggestion should be carefully planned in future experiments.

We hope that all explanations and included changes will be satisfying.

Best Regards

Reviewer 2 Report

This paper portrays in vitro evaluation of the influence of cosmetic substrates and bactericidal nanoparticles, on the viability of two mastitis pathogens.

-        Although there are some post-dip and pre-dip commercial compounds, their environmental effects can be harmful to beneficial soil bacteria due to the long-term persistence of their antibacterial effects. For example, silver nanoparticles have antibacterial properties, but their use has been a cause for concern because they persist in the environment.

-        The title of the manuscript does not fully and accurately indicate the project.

-        Why was the viability of mastitis pathogens + compounds evaluated after 24 hours? Pre-dip compounds should be dried with a towel 30 seconds after use, and in the case of post-dip compounds, they stay on the teat for a maximum of 8 hours and are washed off at the start of a new milking.

-        Although staph aureus and E.coli are important microorganisms that cause mastitis, it would be better to select coagulase-negative staphylococci and corynebacterium, which are more common on the skin of the teat, in addition to staph aureus.

Author Response

Dear Reviewer,

First of all, thank you for valuable suggestions that will help to improve the paper.

According to your comments, I would like to include all necessary changes.

This paper portrays in vitro evaluation of the influence of cosmetic substrates and bactericidal nanoparticles, on the viability of two mastitis pathogens.

-        Although there are some post-dip and pre-dip commercial compounds, their environmental effects can be harmful to beneficial soil bacteria due to the long-term persistence of their antibacterial effects. For example, silver nanoparticles have antibacterial properties, but their use has been a cause for concern because they persist in the environment.

The occurrence and persistence of nanoparticles in environment is not fully described. This problem is mainly observed in water environments and of course this problem should be mentioned in the paper and was added in the Introduction chapter. In addition, authors added also promising studies of positive influence of nanoparticles during embryogenesis (lines 39-50). It is also important issue for the authors for further studies estimating the long-term influence of nanoparticles on dairy cows, especially udder health. The new references were included in the list at the end of the paper:

Szudrowicz, H., Kamaszewski, M., Adamski, A., Skrobisz, M., Frankowska-Łukawska, J., Wójcik, M., ..Bochenek J., Kawalski Kacper, Martynow Jakub., Bujarski P., Pruchniak P., Latoszek E., Bury-Burzymski, Szczepański A., Jaworski S., Matuszewski A., Herman, A. P. (2022). The Effects of Seven-Day Exposure to Silver Nanoparticles on Fertility and Homeostasis of Zebrafish (Danio rerio). International Journal of Molecular Sciences, 23(19), 11239. https://doi.org/10.3390/ijms231911239

Ostaszewska, T., Chojnacki, M., Kamaszewski, M., & Sawosz-Chwalibóg, E. (2016). Histopathological effects of silver and copper nanoparticles on the epidermis, gills, and liver of Siberian sturgeon. Environmental Science and Pollution Research, 23(2), 1621-1633. https://doi.org/10.1007/s11356-015-5391-9.

Matuszewski, A., Łukasiewicz, M., Niemiec, J., Kamaszewski, M., Jaworski, S., Domino, M., Jasiński T., Chwalibóg A., Sawosz, E. (2021). Calcium Carbonate nanoparticles—Toxicity and effect of in ovo inoculation on chicken embryo development, broiler performance and bone status. Animals, 11(4), 932. https://doi.org/10.3390/ani11040932

-        The title of the manuscript does not fully and accurately indicate the project.

The addition of cosmetic substrates is important issue and should be included in the title. Therefore, title was changed according to the received recommendations.

 -        Why was the viability of mastitis pathogens + compounds evaluated after 24 hours? Pre-dip compounds should be dried with a towel 30 seconds after use, and in the case of post-dip compounds, they stay on the teat for a maximum of 8 hours and are washed off at the start of a new milking.

Authors are aware that the milking routine includes 30-40 s contact of pre-dip with teat skin, but this short time would not be efficient for the in vitro study. This issue a  challenge for planned in vivo study. Therefore, authors decided to evaluate how the 24h period of time influences bacteria viability. It was necessary to collect data if selected substrates could positively influence the bacteria viability. If designed mixture left on teat skin had components increasing bacteria multiplication there would be no point of using it in herd conditions. In conclusion, all selected cosmetic substrates should not positively influence bacteria viability and it was main reason to choose 24h period of incubation as the first step of designing the mixture composition. This explanation of chosen methods was also added at the beginning of discussion chapter [147-151].

-        Although staph aureus and E.coli are important microorganisms that cause mastitis, it would be better to select coagulase-negative staphylococci and corynebacterium, which are more common on the skin of the teat, in addition to staph aureus.

The idea of evaluation the effect of designed mixtures on CNS and Corynebacterium sp and other mastitis pathogens eg., fungi, algae, streptococci should be next step of in vitro analysis. Thank you for emphasizing this problem. The authors decided to choose E. coli and S. aureus also because these bacteria are commonly used in different fields of microbiological studies. Moreover, in Poland S.aureus I still common problem in many herds so the high effectiveness of disinfection products against these bacteria is crucial. Thank you for this advice, future research will include higher range of pathogen species. The conclusions were also improved, according to suggestion of the Reviewer.

Best Regards,

The Authors

Round 2

Reviewer 2 Report

The authors' answer is acceptable to me. I hope it will be considered in future studies.